# Association of Livestock Ownership and Household Dietary Quality: Results from a Cross-Sectional Survey from Rural India

**DOI:** 10.3390/ijerph18116060

**Published:** 2021-06-04

**Authors:** Adithya Pradyumna, Mirko S. Winkler, Jürg Utzinger, Andrea Farnham

**Affiliations:** 1Swiss Tropical and Public Health Institute, P.O. Box, CH-4002 Basel, Switzerland; adithya.pradyumna@apu.edu.in (A.P.); juerg.utzinger@swisstph.ch (J.U.); andrea.farnham@swisstph.ch (A.F.); 2University of Basel, P.O. Box, CH-4003 Basel, Switzerland; 3Azim Premji University, Bengaluru 562125, India

**Keywords:** dairy animal, dietary diversity, India, livestock, milk consumption, nutrition

## Abstract

Studies from India and several eastern African countries found that the impact of dairy animal ownership on household nutrition varied greatly, depending on the socio-geographic context. The purpose of this study was to examine the association between livestock ownership and household dietary quality in rural Kolar district, India. We collected data from a household survey in four study villages (*n* = all 195 households of the four villages) of Kolar district, applying a cross-sectional design. Kendall’s rank correlation coefficient was employed to determine the correlation between milk consumption and other dietary variables. Multivariable logistic regression was used to describe the relationship between dairy animal ownership and household milk consumption. Households owning dairy animals more often had access to irrigation (58.3% vs. 25.2%) and were less often woman-headed (2.4% vs. 22.5%). Household milk consumption was significantly correlated with consumption of vegetable variety, egg, and meat (all *p*-values < 0.05). After adjusting for multiple confounders, the odds ratio of milk consumption between dairy animal-owning households as compared to other households was 2.11 (95% confidence interval 0.85, 5.45). While dairy animal ownership was found to be associated with improved dietary quality, larger households were in a better position to adopt dairy animals, which, in turn, might contribute to better household nutrition.

## 1. Introduction

The period 2016–2025 has been declared the Decade of Action on Nutrition by the United Nations [1]. The underlying reason is that child and maternal malnutrition continues to be a major global challenge as a top risk factor for morbidity and mortality worldwide [2], including India [3]. Although considerable progress has been made over the past several years, in 2016 a high prevalence of stunting (38%), wasting (21%), and undernutrition (36%) among children under 5 years of age were recorded in India [4]. Dietary diversity is an important determinant of the nutritional status of children [5,6]. Milk is an important source of animal protein, especially for under 5-year-old children [7,8]. However, analysis of nationally representative data revealed that 5–15% of children in rural areas were at risk of quality protein deficiency, worsening to 26–42% among adults in poor households [9]. Livestock ownership, especially dairy animals, has been promoted in rural India to support livelihood and nutrition [10].

Literature on the impact of dairy animal ownership on diet and nutrition is available from India and several eastern African countries. Livestock ownership was estimated at 59.7% for households across rural settings in India in 2016 [4], and was deemed an important factor in determining consumption of animal-sourced foods [5]. The linkage between dairy animal ownership and milk consumption was specifically emphasized [11]. This was corroborated by studies from Ethiopia [8] and Uganda [12]. However, several factors such as globalization, urbanization, changing access to education, livelihood and progressive reduction in farm sizes are impacting agriculture as a viable livelihood option, complicating its linkages with food security and dietary quality in rural areas, as was found in a study in northern India [7]. Moreover, little is known regarding the association between dairy animal ownership and household milk consumption in villages in southern India.

In a recently conducted qualitative study in Kolar district, a rural semi-arid area in southern India, we found that households that adopted dairy animals for livelihood through the support of watershed development (WSD) projects perceived positive impacts on milk consumption and dietary quality [13]. This pointed to a need for additional quantitative research to deepen the understanding of context-relevant factors, especially keeping in mind the planned promotion of livestock rearing as part of WSD projects in India [14]. Hence, the basis for the current paper was set, and the opportunity for this analysis presented itself when we conducted a cross-sectional baseline survey, as part of a health impact assessment (HIA) of a planned WSD project in four villages of the Kolar district [15]. The specific objectives of this paper were to: (i) describe the status of socio-demographic factors based on dairy animal ownership; (ii) assess the status of health determinants in the study villages based on dairy animal ownership; (iii) determine the correlation of milk consumption with other dietary variables; and (iv) examine the association between livestock ownership and milk consumption in the designated project area.

## 2. Materials and Methods

### 2.1. Study Design and Study Setting

A cross-sectional survey was conducted in the study area during April to July 2019 to characterize socio-demographic factors and health determinants. As the number of people living in each village was relatively small, all households were invited to participate. Data from the cross-sectional survey were used for multivariable logistic regression analysis.

The study was conducted in four neighboring villages in the southern part of Malur sub-district of Kolar district in Karnataka state, India. The total population of these villages was officially enumerated at 1340 individuals in 2011 [16]. The main economic activity is agriculture, with finger millet being the most important crop [16]. The region is drought-prone and vulnerable to climate change [17,18]. The sub-district is in close proximity to Bengaluru, but is largely rural, and has a literacy rate of 62.6%. One out of 10 people (9.4%) belong to scheduled tribes (ST), and one quarter (25.4%) belong to scheduled castes (SC) [19]. In the study villages, 30.8% were of ST and 7.7% were of SC. These villages had access to water, electricity, and sanitation, whereas one village did not have an *anganwadi* (creche). Only 2.1% of households in the project villages reported never consuming meat [15,20]. There was a perception among some households that milk of cattle of high-yielding varieties was unhealthy to consume [13]. Often it was the women of the households who were involved in managing the cattle and milking them. The richer households grew fodder crops, and poorer households used finger millet fodder and also bought fodder to feed the cattle [13], while some pastures were also reportedly available in these villages (6.3% of total land) [16]. Almost all the collected milk was sold to the local dairy, with a little bit for household consumption, or sold to neighbors if they requested [13]. The road connectivity to the nearby town was good, though some of these villages were heavily dependent on private vehicles to access the town [15]. The prevalence of underweight among children was found to be 26.5% in these villages [15,20], which is comparable to the rest of rural Kolar (28.5%, in 2015) [21]. Among 15- to 49-year-old women in rural Kolar, the prevalence of overweight or obesity was 15.2%, and that of anemia was 45.9% [21]. Often one or more members of the household, especially from SC background, traveled outside the village for livelihood and better remuneration [13,15].

These four villages were chosen for the study because a WSD project was planned for the villages by a local non-governmental organization (NGO). These specific villages are geographically part of a micro-watershed, which had been identified for future WSD interventions by local governmental departments. As part of the baseline studies prior to implementing the WSD project, a survey was carried out by our team to deepen the understanding of the social and health situation in the villages, and the relevant data were used for the current research. As several similar villages in this semi-arid region would be subjected to WSD projects, studies of this nature can provide context-specific insights. In addition, the potential of WSD projects to incorporate health aspects such as nutrition was identified by a technical committee in 2006 [22], and hence, such studies can guide nutrition-related activities of WSD projects.

WSD projects have been carried out in semi-arid areas of India by the government, often in partnership with NGOs, aiming to enhance soil and water conservation, improve agricultural productivity, and support livelihoods through approaches such as livestock rearing, especially among the landless households [14]. There are other context relevant interventions conducted as part of these projects by the local project-implementing NGOs, for instance, supporting self-help groups [23]. It was opined by project managers that beneficiaries often opted for high-yielding milch cattle as they were perceived to be a sustainable livelihood source [13].

### 2.2. Data Collection

A structured and pilot-tested questionnaire covering topics of household demography, occupation, agriculture, diet, sanitation, and access to healthcare was used. The questionnaire was administered by trained field enumerators, and data were directly entered into electronic tablets on the Open Data Kit (ODK) platform [24]. Any woman of the household (aged ≥ 15 years) was requested to be the respondent (in some cases this was not possible, so a man became the respondent). Further details of the baseline survey methodology are available elsewhere (see reference [20]).

### 2.3. Statistical Analysis

Data were downloaded from the ODK Aggregate platform and read into R statistical software version 3.5.1 (on RStudio Version 1.1.456) [25]. Variables were summarized to find missing and inappropriate values. Categorical variables were checked for small cell sizes. We examined the associations of socio-demographic factors and other health determinants with ownership of dairy animals—which is the main explanatory variable (binary) for the regression analysis. Dairy animals were defined as high-yielding variety cows, local variety cows, and/or buffaloes. Because all households in the study area were included in the study, confidence intervals (CIs) were not calculated for the descriptive statistics. Kendall’s rank correlation coefficient (tau) was used to calculate the correlation between milk consumption (binary) and other dietary variables (i.e., vegetable consumption, fruit consumption, egg consumption, and meat consumption; each with four ordered categories).

Crude odds ratios (ORs) and 95% CIs were calculated for describing the relationship between household milk consumption (the main outcome; binary variable) and a set of pre-determined covariates (i.e., ownership of dairy animal, household size, sex of household head, existence of children in household, caste (scheduled caste or SC, scheduled tribe or ST and other; with SC and ST considered as marginalized groups in most regions in India), land ownership, access to irrigation, self-help group (SHG) membership and a dummy variable for wage labor as main income source. Household income (self-reported) data were available but not used as a covariate in our analysis because of high potential for measurement errors, and literature from India suggesting that income was an unsuitable predictor of nutrition in rural areas [11]. Several included covariates are proxies for income, for instance, land ownership, access to irrigation, and ownership of motorized vehicles. The covariates were based on a hypothesized causal model of household milk consumption in rural areas in southern India, as ascertained from literature [5,7,8,11,12,26] and knowledge of the local context (Figure 1).

A multivariable logistic regression model adjusting for various covariates was fitted to the data to better understand the causal relationship between dairy animal ownership and household milk consumption. Variables (ownership of other livestock) showing high correlation with the main explanatory variable were eliminated from the final model (Phi coefficient = 0.69). The ORs and 95% CIs were reported and interpreted. Results were considered significant at *p* < 0.05. CIs were reported only for the ORs despite having conducted a census, to provide an impression of the uncertainty around the estimate, especially keeping in mind the small population size, to aid in generalizability of study results outside of the study population.

Sensitivity analysis was performed by using propensity score matching (PSM) through nearest neighbor matching (NNM) with replacement, using the MatchIt package in R statistical software, to increase the comparability of the two groups based on the following covariates: household size, sex of household head, child in household, caste, land ownership, access to irrigation, SHG membership, and wage labor. The quality of matching was assessed through quantile-quantile (QQ) plots and two sample *t*-tests for each covariate. As the matching was deemed to have improved comparability (though having expectedly reduced the sample size), multivariate logistic regression was carried out on this data subset to better understand the association between dairy animal ownership and household milk consumption.

## 3. Results

### 3.1. Socio-Demographic Characteristics of the Study Population

A total of 195 households were included in the four project villages (response rate: 100%; this includes all households of the project villages that were available in the village during the survey period). Over 93% of the respondents were adult women. The median age of all respondents was 35 years (25th–75th percentile: 27–45 years). Most of the respondents were illiterate (56.9%). Characteristics of the respondents are summarized in Table 1. Less than half of the households (43.1%; *n* = 84) owned at least one dairy animal. Households owning dairy animal(s) were more often larger than households not owning a dairy animal (median of 5 persons vs. 4), had greater land-holdings (2.75 acres vs. 2.09 acres), increased access to irrigation (58.3% vs. 25.2%), more frequent ownership of a motorized vehicle (92.9% vs. 81.1%) and higher ownership of other livestock (95.2% vs. 26.1%). Households owning dairy animal(s) were also less likely to be woman-headed (2.4% vs. 22.5%), SC (2.4% vs. 11.7%), or have recent history of seasonal migration (2.4% against 9.0%).

Details on ownership of livestock in the study population are shown in Figure 2. Those owning dairy animals (local variety cows, buffaloes, and/or high-yielding variety cows) more often had greater variety of livestock (including chicken, goats, sheep, oxen, and/or calves).

### 3.2. Status of Health Determinants and Correlation of Household Milk Consumption with Dietary Variables

The status of selected health determinants is summarized in Table 2. One in five households (20.5%) have experienced food insecurity during the past two years. A higher proportion of households owning dairy animal(s) consumed milk (88.6% vs. 62.2%). Latrine ownership was high across both groups (92.8%). Dairy animal-owning households more commonly opted for private healthcare services in case a household member had fever (38.1% vs. 15.3%). Health insurance coverage was only reported by 44.1% of households. Milk consumption was significantly correlated with vegetable consumption (Kendall’s tau = 0.16, *p* = 0.017), egg consumption (Kendall’s tau = 0.27, *p* < 0.001), and meat consumption (Kendall’s tau = 0.14, *p* = 0.049). The correlation statistics are summarized in Table 3.

### 3.3. Association between Ownership of Dairy Animal(s) and Milk Consumption

Of the 195 households, 52 (26.7%) reported that they did not consume milk. Covariates that showed strong crude associations with milk consumption include owning dairy animal(s) (OR: 4.50, 95% CI: 2.17, 10.14), household size (OR: 2.00, 95% CI: 1.56, 2.66), woman-headed households (OR: 0.19, 95% CI: 0.08, 0.43), land ownership (OR: 1.67, 95% CI: 1.23, 2.35), access to irrigation (OR: 3.19, 95% CI: 1.57, 6.99), and owning a motorized vehicle (OR: 9.72, 95% CI: 4.04, 25.4) (Table 4).

The full model for the relationship between dairy cow ownership (primary explanatory variable) and milk consumption (main outcome) was adjusted by household size (count), woman-headed household (binary), whether general caste (binary), child in household (binary), wage labor as main income source (binary), land owned (continuous), access to irrigation (binary), membership in SHG (binary), and ownership of motorized vehicle (binary). The multivariate logistic regression model output indicated that the adjusted OR for household milk consumption was 2.11 (95% CI: 0.87, 5.45) between households owning and not owning dairy animals. Evidence of association was found for household size (adjusted OR: 1.88, 95% CI: 1.34, 2.77), ownership of motorized vehicle (adjusted OR: 4.08, 95% CI: 1.23, 14.31) and wage labor as primary income source for family (adjusted OR: 2.89, 95% CI: 1.04, 9.03).

The sensitivity analysis included 128 households (84 households owning dairy animals and 44 not owning dairy animals). Previously observed imbalances between households owning and not owning dairy animals were minimized through the matching for all included covariates, except for household size (difference persisted after matching at *p* = 0.03). The adjusted OR for milk consumption was 2.20 (95% CI: 0.77, 6.45) in this subsample.

## 4. Discussion

Milk consumption was found to be both significantly correlated with other markers of a diverse and high-quality diet (i.e., vegetable, egg, and meat consumption) and elevated among households owning dairy animal(s) (OR: 2.11), even after controlling for multiple confounders. The association between dairy animal ownership and milk consumption was not statistically significant after adjusting for all covariates (see the column “Adjusted OR” in Table 4), which might be explained by the relatively small sample size (n = 195). Of note, we included all households in the four villages that will be affected by the project. It is conceivable that families owning dairy animals would consume milk when the dairy animal is producing milk. However, families owning dairy animals were, in general, different from those not owning dairy animals. For example, they often owned greater assets (land, access to irrigation, and motorized vehicle(s)) than those that did not. This was also consistent with the finding that woman-headed and SC households owned dairy animals far less frequently. This suggests ownership of dairy animals was associated with an overall higher socioeconomic status in the study area. The investment of purchasing and managing a high-yielding dairy animal may be prohibitive to those without assets [7,12]. However, even after matching for these socioeconomic covariates through PSM, a positive association was found between dairy cow ownership and milk consumption, suggesting that this finding is not entirely dependent on socioeconomic status.

The advantage of owning land and irrigation access for meeting fodder and water needs of dairy animals was reported by other studies [7,12]. Indeed, some local farmers from nearby villages revealed cultivating only fodder crops in their irrigated fields, focusing solely on dairy for livelihood [13]. This indicates that while dairy animals have the potential to contribute dietary quality and diversity, the impact may be disproportionately higher for richer households, as has been shown in an earlier study [12].

### 4.1. Interpretation from a Household Nutrition Perspective

The main finding of an association between dairy animal ownership and household milk consumption was corroborated by a large study from India [11], and also smaller studies from Ethiopia (23% increased frequency) [8], Uganda [12,27], and Kenya [28]. This was found to be especially important in areas without access to markets [8], which was not the case in our study area where dairies have been established.

While milk was significantly correlated with other markers of a diverse and high quality diet, it was not the only source of protein and micronutrients in the study area, as has also been reported in literature [9]. There is also consumption of finger millet, pulses (a regular feature in meals), eggs and meat, the latter two being more frequent among households owning dairy animals. These findings are in contrast to what was observed in some villages in northern India where consumption of milk and milk products were found to be more critical to dietary quality [7]. The importance of understanding local context in the contribution towards household nutrition is emphasized [7].

The proportion of households that reported having experienced food insecurity during the last two years was similar for households with or without dairy animal(s) (17.9% vs. 22.5%). These percentages do not indicate the frequency and severity of the experienced food insecurity. In addition, it is difficult to draw causal inferences in the context of dairy animal ownership as this is a cross-sectional study.

Food consumption at the household level cannot be extrapolated to nutritional status of individuals within the household, as shown before [6,11]. A study from Ethiopia indicated positive impact on reducing stunting [8]. Studies from Uganda found significant positive impact [12], no impact [27], or even negative impact [26] of dairy animal ownership on child nutrition, and hence, there must be other contextual factors, such as availability and use of sanitation and intra-household competition for resources. Small ruminants (e.g., goats and sheep) were found to contribute to better nutrition outcomes in Uganda [26] and the poorest households in Kenya [29]. Several other factors complicating this relationship have been elucidated in the literature, including wealth, resource constraints, and experience of financial shocks [8].

### 4.2. Interpretation in the Light of WSD Projects

WSD projects locally have helped overcome the obstacle of high initial investment by providing grants and loans to procure livestock, preferentially to poor woman-headed households through SHGs [13]. Currently SHG membership was somewhat lower among households without dairy animals (33.3% against 41.7%), and this can be expected to improve through the planned WSD project [15]. Beneficiaries in earlier local WSD projects perceived financial and nutritional benefits following the adoption of dairy animal(s) [13]. On similar lines, an intervention study in Rwanda on donation of livestock to households was able to demonstrate impact on child nutrition [30]. However, keeping in mind that managing dairy animals is labor-intensive and harbors various costs [31,32]—including accessing water and feed [7], all households may not be able to adopt it. The varying success of dairy programs in villages in northern India due to the role of availability of land and labor in the household has also been reported [33].

Interventions encouraging dairy animal ownership as part of the WSD project should take into account whether it is feasible for low-income households to maintain a dairy animal long-term. Additionally, challenges of water and feed are worsened during droughts [7], which occur regularly in Kolar district. Local anecdotal evidence (assimilated during a recent study [13]) reported that several households sold their dairy animals a few years ago following a period of intense drought. Financial returns from dairy animals were also reportedly lower in areas with high groundwater exploitation [32], such as in the study area. In Tamil Nadu, an “economically transforming” state, it was also observed that smallholder farmers had downsized dairy farming in the 10 years prior to the 2018 study for various economic and cultural reasons [34], and this may have bearing for Kolar, which is a neighboring district of Tamil Nadu. In addition, promotion of dairy animals comes with health and ethical challenges such as antimicrobial resistance, especially for high-yielding varieties [35]. Therefore, this strategy could be reviewed accordingly.

### 4.3. Interpreting Effects of Household Size and Wage Labour—Novel Results

We found a strong association of milk consumption and household size (OR: 1.88, 95% CI: 1.34, 2.77) (Table 4), which is in contrast to findings from a large Indian dataset [11]. Two factors might explain this observation. First, wealthy households in the study area lived as joint families, as they have the financial and human resources to buy and manage dairy animals. Second, the poorest households were those of elderly women living alone. The strong association between wage labor and milk consumption (OR: 2.89, 95% CI 1.04, 9.03) may also be related to few households consisting only of elderly poor women living alone unable to engage in wage labor. Reportedly, regular wage labor in construction industry and domestic work in nearby cities was providing adequate returns to young people from this area [13].

### 4.4. Limitations of the Study

It is not possible to draw conclusions on causal relationships from cross-sectional data. Reverse causality between household milk consumption and dairy animal ownership is plausible if milk consumption is considered a proxy for wealth/income. Keeping the literature and context in mind, this is unlikely. However, as ownership of cattle was strongly associated with wealth indicators, the association with household milk consumption should be interpreted with caution. Another limitation of the analysis was the lack of data on other milk products. In our preceding work in the same region, we found that part of the milk was consumed in fermented form (curd) [13]. As this curd was made from fresh milk within the household, we assumed it was represented within the data on milk consumption. Finally, the findings of the present study mainly apply to the study area, but may also provide insights on what can be expected in the drought-prone rural regions in southern India.

### 4.5. Scope for Future Research and Practice

Further research could adopt a prospective mixed-method design, focus on differential benefits experienced by various types of adopting households, and also study challenges being faced by each in taking up and managing dairy animals. Adding outcome measures (e.g., nutritional status among children within the household, hemoglobin levels among adults) as part of the survey would be good to indicate the size and distribution of direct health impacts of these interventions. This kind of evidence is currently lacking [5]. The experience from Gujarat also indicates the need to consider larger economic aspects and cultural dynamics in dairy promotion [36], and such studies with social science perspectives need to be conducted in Kolar’s context.

Keeping in mind that the data for the study came from a baseline survey, the additional benefits of conducting comprehensive health impact assessments for agricultural projects was revealed—fostering empirical research in neglected settings [37]. Indeed, baseline survey data can be leveraged to better understand agriculture and nutrition linkages, besides other locally relevant health outcomes.

## 5. Conclusions

Our study revealed that dairy animal ownership was quite common in the study area (55.9% of the households, as compared to 59.7% for rural India on average [4]). We found evidence suggestive of causal relationship between dairy animal ownership and household milk consumption in the four villages in the southern part of Kolar district. Households consuming milk were found to have a better dietary quality in terms of vegetable variety, frequency of meat consumption and frequency of egg consumption. In terms of the factors associated with adoption of dairy animals, we found that wealth, household size, land ownership and access to irrigation were important. Our findings also illustrated how context plays a role in determining effects of interventions in rural areas, for instance, the effect of household size.

Health- and equity-sensitive rural development schemes are needed for achieving Sustainable Development Goals 3 (good health and well-being) and 10 (reduce inequalities). More specifically, a call has been made for development policies to be nutrition-sensitive [38,39]. Both WSD projects and the livestock mission have been recognized for the potential to also address nutritional challenges. Based on our findings and the literature, we recommend that, while there is merit for continued support for livestock programs, there is a need for incorporating contextual insights into program design to ensure that it is relevant and to have realistic expectations of returns in the form of better nutrition and/or improved livelihood. This emphasizes the role of district offices, local organizations and research institutions in the process. The type of livestock would also be an important factor. As considerable resources are put into these initiatives, careful monitoring and evaluation of these interventions and schemes is essential, with periodic revision of interventions based on key findings. One-size-fits-all approaches cannot be expected for diverse contexts.

We used the opportunity of a baseline survey of a planned project to contribute to literature and local planning. Further social science-oriented studies can provide further insights on the utility and appropriateness of livestock interventions for improving nutrition. Similar studies from other parts of the country could also further enhance our understanding about agriculture–nutrition interlinkages towards addressing undernutrition.

## Figures and Tables

**Figure 1 ijerph-18-06060-f001:**
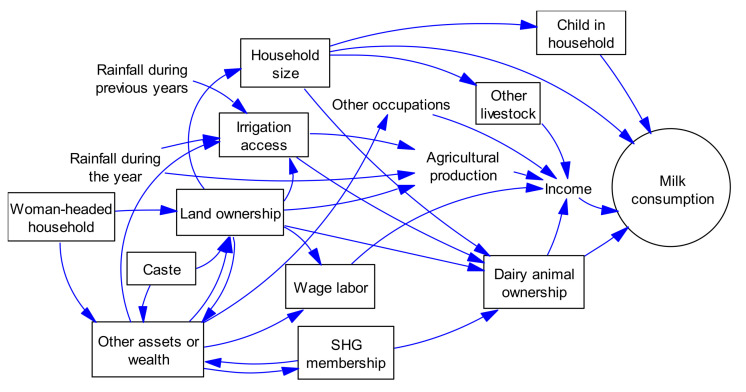
Factors potentially influencing milk consumption at household level in the study area (boxed variables have been included in the analysis); SHG, self-help group.

**Figure 2 ijerph-18-06060-f002:**
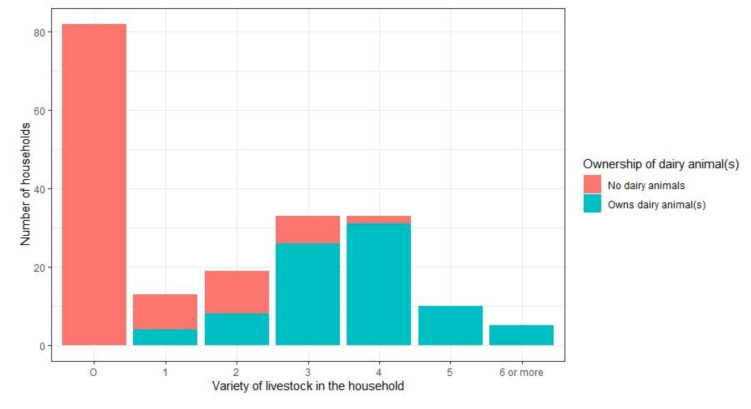
Ownership of livestock in the study population based on a survey conducted between April and July 2019 in four villages in Kolar district, India.

**Table 1 ijerph-18-06060-t001:** Socio-demographic characteristics of the study population from a household survey conducted between April and July 2019 in four villages in Kolar district, India.

Variable	Owns Dairy Animal(s) (*n* = 84)	Does Not Own Dairy Animal(s) (*n* = 111)	Total (*n* = 195)
Respondent details			
Age (median [P25–75]) ^a^ in years	34.5 (26.0–41.3)	35 (28–47)	35 (27–45)
Respondent is female	75 (89.3%)	107 (96.4%)	182 (93.3%)
Respondent is illiterate	42 (50.0%)	69 (62.2%)	111 (56.9%)
Household characteristics			
Household size (median (P25–75))	5 (4.8–7)	4 (3–5)	5 (4–6)
Under-5 child in household	23 (27.4%)	22 (19.8%)	45 (23.1%)
Woman-headed household	2 (2.4%)	25 (22.5%)	27 (13.8%)
Caste			
*General category*	52 (61.9%)	68 (61.3%)	120 (61.5%)
*Scheduled caste (SC)*	2 (2.4%)	13 (11.7%)	15 (7.7%)
*Scheduled tribe (ST)*	30 (35.7%)	30 (27.0%)	60 (30.8%)
Primary income source			
*Agriculture*	61 (72.6%)	70 (63.1%)	131 (67.2%)
*Daily wage*	14 (16.7%)	24 (21.6%)	38 (19.5%)
*Livestock rearing*	3 (3.6%)	2 (1.8%)	5 (2.6%)
*Other*	6 (7.2%)	15 (13.5%)	21 (10.8%)
Land ownership in acres ^b^ (mean [standard deviation])	2.75 [1.47]	2.09 [1.46]	2.37 [1.49]
Access to irrigation	49 (58.3%)	28 (25.2%)	77 (39.5%)
Owns non-dairy livestock	80 (95.2%)	29 (26.1%)	109 (55.9%)
Regular travel for wage labor	33 (39.3%)	49 (44.1%)	82 (42.1%)
Undertook seasonal migration	2 (2.4%)	10 (9.0%)	12 (6.2%)
SHG membership	35 (41.7%)	37 (33.3%)	72 (36.9%)
Owning a motorizedvehicle	78 (92.9%)	90 (81.1%)	168 (86.2%)
Social welfare card			
*Below poverty line*	77 (91.7%)	105 (94.6%)	182 (93.3%)
*Antyodaya scheme*	7 (8.3%)	5 (4.5%)	12 (6.2%)

^a^ 25th and 75th percentile; ^b^ one acre = 4046.86 m^2^.

**Table 2 ijerph-18-06060-t002:** Select health determinants in the study population based on a survey conducted between April and July 2019 in four villages in Kolar district, India.

Variable	Owns Dairy Animal(s) (*n* = 84)	Does Not Own Dairy Animal (*n* = 111)	Total (*n* = 195)
Experienced food insecurity in the past two years	15 (17.9%)	25 (22.5%)	40 (20.5%)
Consume any milk regularly	74 (88.1%)	69 (62.2%)	143 (73.3%)
Variety of vegetables consumed previous week (median (P25–P75) ^a^	6 (5–7)	6 (4–7)	6 (5–7)
Egg consumption frequency in a month (median (P25–P75)	2 (2–4)	2 (1–3)	2 (1–4)
Meat consumption frequency in a month	4 (4–4.3)	4 (3–4)	4 (4–4)
Fruit consumption frequency in a month (median (P25–P75)	2 (1–3)	1 (0–2)	1 (0–2)
No knowledge of any iron-rich foods	2 (2.4%)	11 (9.9%)	13 (6.7%)
Latrine ownership	79 (94.0%)	102 (91.9%)	181 (92.8%)
Any member consumesalcohol	7 (8.3%)	20 (18.0%)	27 (13.8%)
Any member smokes	11 (13.1%)	20 (18.0%)	31 (15.9%)
Any member chews tobacco	21 (25.0%)	25 (22.5%)	46 (23.6%)
First choice healthcare provider for fever			
*Local government hospital*	50 (59.5%)	91 (82.0%)	141 (72.3%)
*Local private doctor*	32 (38.1%)	17 (15.3%)	49 (25.1%)
Health insurance cover			
*Governmental schemes*	34 (40.5%)	51 (45.9%)	85 (43.6%)
*Private*	1 (1.2%)	0 (0.0%)	1 (0.5%)
*None*	49 (58.3%)	60 (54.1%)	109 (55.9%)

^a^ 25th and 75th percentile.

**Table 3 ijerph-18-06060-t003:** Correlation of milk consumption with other dietary variables based on data from survey conducted between April and July 2019 in four villages in Kolar district, India.

Factor Related to Dietary Quality	Kendall’s tau	*p*-Value
Variety of vegetables consumed	0.163	0.017 *
Frequency of fruit consumption	0.016	0.816
Frequency of egg consumption	0.265	<0.001 *
Frequency of meat consumption	0.139	0.049 *

* Significant at *p*-value < 0.05.

**Table 4 ijerph-18-06060-t004:** Crude and adjusted odds ratios (ORs) and a sensitivity analysis (SA) comparing household milk consumption with the explanatory variables based on data collected between April and July 2019 from four villages in Kolar district, India.

Variable	Crude OR (95% CI)	Adjusted OR (95% CI)	SA: Adjusted OR (95% CI)
Owns dairy animal(s)	4.50 (2.17–10.14) ***	2.11 (0.87–5.45)	2.20 (0.77–6.45)
Household size	2.00 (1.56–2.66) ***	1.88 (1.34–2.77) ***	1.62 (1.06–2.77)
Woman-headed household	0.19 (0.08–0.43) ***	0.78 (0.25–2.58)	NA
Whether SC ^a^	2.50 (0.66–16.35)	-	-
Whether ST ^a^	1.13 (0.57–2.32)	-	-
Whether general caste	0.71 (0.36–1.37)	0.71 (0.31–1.58)	0.36 (0.1–1.14)
Child in household	1.92 (0.86–4.73)	0.48 (0.16–1.45)	0.73 (0.15–4.16)
Wage labor main income source	0.87 (0.4–1.97)	2.89 (1.04–9.03)	2.01 (0.52–9.17)
Land owned	1.67 (1.23–2.35) **	1.06 (0.8–1.47)	1.8 (0.96–3.92)
Irrigation access	3.19 (1.57–6.99) **	1.30 (0.53–3.29)	1.75 (0.55–5.74)
SHG member	1.29 (0.67–2.56)	1.04 (0.47–2.37)	3.38 (0.97–14.98)
Owns motorised vehicle	9.72 (4.04–25.4) ***	4.08 (1.23–14.31) *	1.21 (0.19–6.53)
Any non-dairy livestock owned ^b^	3.71 (1.92–7.42) ***	-	-

*** *p*-value < 0.001; ** *p*-value < 0.01; * *p*-value < 0.05; ^a^ due to small cell sizes, only a dummy variable for general caste was used in the final model; ^b^ highly correlated with ownership of dairy animal, hence not included in final model; NA, not available; SA, sensitivity analysis with data subset determined by propensity score matching; SC, scheduled caste; SHG, self-help group; ST, scheduled tribe.

## Data Availability

Data is available upon request to the author.

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
