# Peer review of "Association of Livestock Ownership and Household Dietary Quality: Results from a Cross-Sectional Survey from Rural India"

_ijerph, 2021, doi:10.3390/ijerph18116060_

Round 1

Reviewer 1 Report

This is a very useful study on the association between nutrition and dairy animal ownership in south India. It is well written and organized, and provides valuable insights on village-level conditions. However, there are some issues that could be considered in order to further clarify this study.

-- 'households that adopted dairy animals for livelihood through the support of watershed development (WSD) projects' [lines 53-54]: Need some more background on the WSD program and why (high yielding? improved?) dairy animals were being provided. What else does the program provide? This can be mentioned in Section 2.1.

-- In Section 2.1, need to provide some justification on why the four villages were chosen as study sites. Why do these villages and Malur sub-district become useful to understanding the relationship between milk consumption and dairy animal ownership? (For instance, is it because dairy animals were recently introduced? Because WSD has a special focus on nutrition?) Need for studies on south India is a useful justification, but does not adequately explain why the specific study sites are valuable.

-- There needs to be some more information on the four villages, or Kolar sub-district.

In terms of the sub-district, what is its literacy, sex ratio, quality of housing, access to electricity, piped water, toilet within the house, extent of urbanization, any other variables authors view as pertinent?

In terms of the villages, some more detail on the dairy economy and the wider rural economy would be useful: are there any caste taboos related to food, is there any common land for grazing, who does the work of milking [women, hired help], is milk sold in the community or nearby markets [line 234, dairies have been established], is milk processed for household use [there is mention of curd, lines 297-301], what is the quality of road connection to nearby town? Any other aspects that the authors consider pertinent to understanding the broader context of nutrition availability should also be mentioned.

Establishing the unique aspects of the study sites would be useful in terms of comparing this study to other parts of India, and thus increasing its generalizability.

-- Geographic coordinates need not be mentioned [lines 74-75]?

-- Section 2 needs to discuss the variables utilized in a separate section, maybe after data collection. There seem to be three different sets of variables: socio-demographic characteristics, health determinants, and explanatory variables (with some overlap between socio-demographic characteristics and explanatory variables). What explains this division of variables?

Need to provide a descriptive statistics table which lists all variables and their mean, range, etc. This is where the different variable categories (socio-demographic, health) can be clarified.

-- 'We examined the associations of socio-demographic factors and health determinants with ownership of dairy animals; the main explanatory variable, defined as high-yielding variety cows, local variety cows and buffaloes.' [lines 91-93]

This seems to be two sentences. The main explanatory variable should be detailed in a separate sentence.

-- Were the 195 households the total of all households in the four villages? On lines 210-211, it is mentioned that '... we included all households in the four villages that will be affected by the project:' Not clear if all households affected by project is the same as all households per se?

Would be useful to reiterate that all households were included at the beginning of section 3.1.

-- 'The association between dairy animal ownership and milk consumption was not statistically significant, which might be explained by the relatively small sample size (n=195). [lines 208-210]

This needs some more explanation. Where is this association depicted? Table 4 seems to suggest that this association is significant?

-- Figure 2 needs to be explained in some more detail, either in Results or Discussion. The figure shows that households with dairy animals often also own a variety of livestock species. What are these livestock species?

Are there any caste taboos in terms of consumption of meat? In some other parts of India, a taboo on meat eating means that a variety of livestock species is usually associated with households of specific castes. Would be useful to get some insight on how caste based traditions related to food or occupation shape livestock ownership  in terms of the study sites.

-- Section 4.3 on effects of household size and wage labor can be shifted to Results [Section 3.3], or should also be mentioned in Results. Alternatively, section 4.3. can be titled 'Novel results'?

-- 'Our study revealed that dairy animal ownership was quite common in the study area, but lower than the average for rural India.' [lines 318-319] Lower than average animal ownership compared to rest of rural India needs a reference. Or should be deleted, since the study does not focus on a comparison with the rest of India.

-- line 313: provide full form for HIA

-- Abstract:

line 10: 'the government:' need to specify whether this is governments across the world, or a specific government. Need some example of programs here? Or delete the first sentence?

line 14: 'exhaustive:' not clear what this means. Delete this term or use a different descriptor.

-- Authors might consider the following references if pertinent to their study:

-- Thirunavukkarasu, Duraisamy, Narmatha, Natchiappa, Doraisamy, Kolathur Arumugam, Saravanakumar, Velusamy Ramesh, Sakthivel, Kokumadai Mottaiappan (2019). Future Prospects of Smallholder Dairy Production: Pragmatic Evidence from Crop-Livestock Farming Systems of an Economically Transforming State in India. Cuadernos de Desarrollo Rural, 16(84).

-- Daftary, D. (2019). Market-driven dairying and the politics of value, labor and affect in Gujarat, India. Journal of Peasant Studies, 46(1), 80-95.

-- Daftary, D. (2018). Cattle, milk and women’s labour: The politics of contemporary dairying in Gujarat. Economic & Political Weekly, 53(22): 43-50.

-- Pratyusha Basu & Jayajit Chakraborty (2008) Land, Labor, and Rural Development: Analyzing Participation in India's Village Dairy Cooperatives, The Professional Geographer, 60:3, 299-313, DOI: 10.1080/00330120801985729

Reviewer 2 Report

Technical comments and recommendations

I like the information presented in figure 1. However, This graphic would be much more meaningful and analytical if it presented the converse information also, that is, factors that resulted in low milk consumption. Alternatively, and perhaps better, would be to provide a similar graphic in the results or conclusion section of the paper showing the factors associated with lower milk consumption and lower ownership of dairy animals.

The first line of the abstract (10-11) states, "Livestock rearing is being promoted by the government to improve household income and nutrition." Based on this statement, I exppected that the paper would examine factors necessary to ensure that any program implemented by the government to "improve household income and nutrition" would be successful. Researchers provided some excellent information in the 4. Discussion section of this paper on the factors that inhibit various sectors of the population from being able to obtain or continue to maintain dairy animals. They also discussed other sources of protein that can fulfill the nutritional needs of people who drink less milk. Finally, the authors provided an excellent discussion of additional factors to consider in future research so as to develop more precise recommendations for restructuring government programs to enhance the nutrition of the resource-poor sectors of society. 

Based on the quality and type of information developed through this research, I found the conclusion section to be less informative or much more restricted in its recommendations than is consistent with the remainder of the paper. I believe that this well written research paper could be made better by discussing what might be factors that government agencies might consider incorporating into future government programs while awaiting additional research information.

Based on these observations, 

 the government should consider addressing to make a program to enhance dairy animal ownership successful. 

Reviewer 3 Report

Dear author(s),

The topic developed is of interest and relevance and it offers several references to support it. I’ve recommended that your paper is suitable for publication in IJERPH. I am completely sure of your ability to make progress on this subject. I encourage you to carry on down the path you have chosen.

Round 2

Reviewer 1 Report

Thank you for incorporating the suggestions and revising the article. A few minor issues remain that are listed below and detailed under Comments in the attached pdf file:

-- In the first line of the Abstract [line 10], replace 'Africa' with 'several African countries' or 'East Africa.'

-- lines 60-64: The objectives listed here could follow the order of the Results, so that health determinants and correlation could be (ii). Also, 'are' needs to be inserted after 'specific objectives of this paper ...'

-- The sub-section titles under Results could be revised. Title for 3.1 could be 'Socio-demographic characteristics of study population,' and title for Section 3.2 could be 'Health determinants of study population and correlation between milk consumption and other sources of nutrition.'

Also, use either 'other sources of nutrition' or 'other dietary variables' throughout.

-- Delete 'based on survey conducted between April and July 2019 in four villages in Kolar district, India' from all figures and tables, or add this statement as source below figures/tables.

-- Delete first sentence of Section 3.2 [lines 198-199], and begin with 'The status of health determinants is summarised ...'

I now think that it is fine to call the second set of variables 'health determinants,' so please delete 'other' throughout the text.

-- In Table 2, it could be either 'Monthly frequency of egg consumption' or 'Egg consumption-monthly frequency.' Make this consistent for all three variables.

Additional minor changes are listed in attached file.
